

# The crossdip correction as a tool to improve imaging of crooked line seismic data: A case study from the post-glacial Burträsk fault, Sweden

Ruth A. Beckel[1] and Christopher Juhlin[1]

[1]Department of Earth Sciences, Uppsala University, Sweden

**Correspondence:** R. Beckel (ruth.beckel@geo.uu.se)

**Abstract.**

Understanding the development of post-glacial faults and their associated seismic activity is crucial for risk assessment in Scandinavia. However, imaging these features and their geological environment is complicated due to special challenges of their hardrock setting, such as weak impedance contrasts, sometimes high noise levels and crooked acquisition lines. A crooked line geometry can cause time shifts that seriously de-focus and deform reflections containing a crossdip component. Advanced processing methods like swath 3D processing and 3D pre-stack migration can, in principle, handle the crooked line geometry, but may fail when the noise level is too high. For these cases, the effects of reflector crossdip can be compensated for by introducing a linear correction term into the standard processing flow. However, existing implementations of the crossdip correction rely on a slant stack approach which can, for some geometries, lead to a duplication of reflections. Here we present a module for the crossdip correction that avoids the reflection duplication problem by shifting the reflections prior to stacking. Based on tests with synthetic data, we developed an iterative processing scheme where a sequence consisting of crossdip correction, velocity analysis and DMO correction is repeated until the stacked image converges. Using our new module to reprocess a reflection seismic profile over the post-glacial Burträsk Fault in Northern Sweden increased the image quality significantly. Strike and dip information extracted from the crossdip analysis helped to interpret a set of southeast dipping reflections as shear zones belonging to the regional scale Burträsk Shear Zone (BSZ), implying that the BSZ itself is not a vertical, but a southeast dipping feature. Our results demonstrate that the crossdip correction is a highly useful alternative to more sophisticated processing methods for noisy datasets. This highlights the often underestimated potential of rather simple, but noise-tolerant methods, in processing hardrock seismic data.

## 1 Introduction

Today, northern Scandinavia is generally considered to be a low seismic hazard area. There are a number of intraplate earthquakes connected to the post-glacial rebound, but only very few of them exceeding a magnitude of 4. In the past, however, the post-glacial adjustments seem to have been more intense. Throughout northern Scandinavia, up to 15 m high fault scarps extending for tens of kilometers (Fig. 1) suggest the occurrence of violent earthquakes at the end or directly after the last glacial retreat (e.g. Lagerbäck and Sundh, 2008; Olesen et al., 2013; Kuivamäki et al., 1998). Based on sediment deformation



and liquefaction features, the magnitude of the earthquakes associated with some of these post- or end-glacial faults has been estimated to be in the order of magnitude of 7-8 (Arvidsson, 1996; Mörner, 2005). Since the discovery of the first faults in the late 1980s, they have been of special interest to scientists and society. The most important question is, of course, if the large intraplate earthquakes are repeatable or unique events.

Understanding the post-glacial faults requires imaging their deeper structure – which is a challenge of its own. On the one hand, finding a balance between signal penetration and resolution to be able to image a (potentially) very narrow shear zone at several kilometers depth is required. On the other hand, inaccessibility of the terrain in northern Scandinavia confines seismic data acquisition to existing roads and tracks, often resulting in profiles with a very crooked geometry. If a reflector has a crossdip component, this can lead to focusing problems and time shifts that significantly reduce the quality of the stacked

image.

These challenges are not unique to post-glacial faults and several different processing approaches exist. Among the possible approaches are advanced methods like 3D processing and 3D pre-stack migration. Although treating a swath 3D survey over a 3D geological structure as a proper 3D dataset is doubtlessly the most appropriate approach, there are some limitations due to the small crossline aperture and uneven midpoint distributions (Nedimović and West, 2003). Moreover, binning the traces in

3D reduces the average data fold considerably. In typical hardrock settings, often exhibiting a comparably low signal-to-noise level, this might affect the quality of the final image considerably. A more simplistic, but more noise-tolerant, approach is to correct for the effects of crossdipping reflectors and continue processing the dataset in 2D. This idea is not new (e.g. Larner et al., 1979; Du Bois et al., 1990; Kim and Moon, 1992; Nedimović and West, 2003), but none of the existing correction methods is optimally suited to image a feature like a post-glacial fault. Thus, we decided to adapt the crossdip correction to

our needs and develop a module that is easy to use within our processing software. Moreover, the implications of the crossdip correction have never been tested extensively. Possible interactions with other processing steps, like the dip-moveout (DMO) correction, should especially be studied in more detail.

Since 2007, a couple of reflection seismic profiles over post-glacial faults have been recorded. In case of the Pärvie (Juhlin et al., 2010; Ahmadi et al., 2014) and Suasselkä faults (Kukkonen et al., 2009; Abdi et al., 2015), the fault scarps could be

linked with dipping reflections but for the Burträsk fault (Juhlin and Lund, 2011), no reflection directly connected to the fault scarp was observed. However, Juhlin and Lund (2011) imaged a dipping reflection which they could indirectly link to the fault. They attributed the lack of a clear reflection at the fault scarp to the crookedness of the profile and the complex geometry of the fault at the intersection with the seismic line (Fig. 2). The Burträsk fault is, however, of special interest since it is the seismically most active post-glacial fault and one of the most seismically active areas in the whole of Sweden. Therefore, we

decided to reprocess the Burträsk dataset, applying an adapted crossdip correction to the data.

This paper is divided into two parts. The objectives of the first part are to develop a local crossdip correction module and to test its interactions with other processing steps to establish an optimized processing scheme. In the second part, we apply our crossdip correction module to the Burträsk dataset to improve both the imaging of the fault and other reflectors below the profile and to refine the geological interpretation using information about the strike and dip of the reflections obtained by the

crossdip analysis.



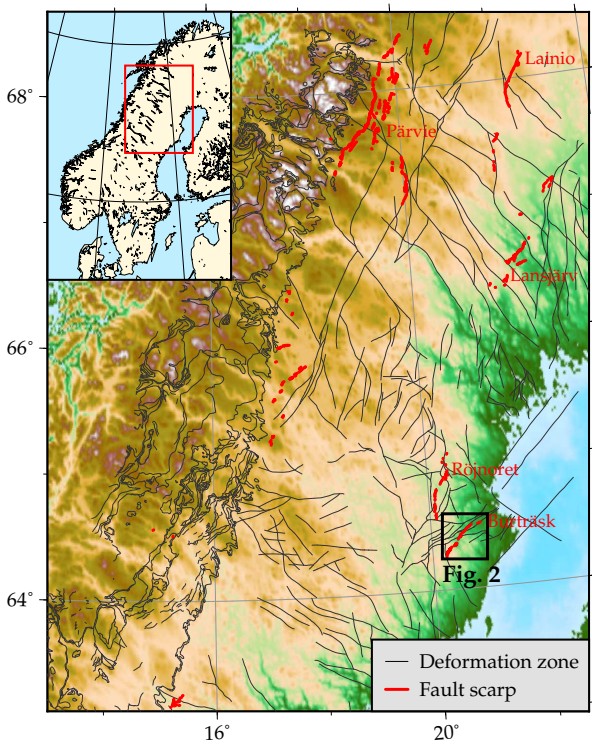

**Figure 1.** Post-glacial fault scarps in northern Sweden derived from LiDAR data (Mikko et al., 2015). The majority of the post-glacial faults strike north-northeast or northeast but both the Burträsk and Röjnoret faults deviate significantly from this trend. Deformation zones ©Geological Survey of Sweden.

## 2   Geological setting

The survey area is situated in the Paleoproterozoic rock formations of the southern Skellefteå District. The main structural feature of the area is a wide, dextral shear zone, suggested to have formed by lateral escape during the Svecokarelian orogeny (Romer and Nisca, 1995). Following Romer and Nisca (1995), we will refer to this feature as the Burträsk Shear Zone (BSZ).

5   It was subdivided further by Rutland et al. (2001), but for simplicity we continue using the original definition.

The BSZ marks the transition from metasedimentary rocks in the south to an area dominated by magmatic rocks in the north (Fig. 2). The metasedimentary rocks belong to the Bothnian supergroup – a sequence of sediments of mostly turbiditic origin that was accreted and metamorphosed during the Svecokarelian orogeny. They consist mainly of highly deformed and migmatized meta-graywackes, meta-argillites and paragneisses (Kathol and Weihed, 2005). The magmatic rocks in the northern

10   part are attributed to have originated from two different phases of magmatism: The early Svecokarelian calc-alkaline intrusive rocks are mostly of granodioritc or tonalitic composition and are tentatively dated to 1.96-1.86 Ga (Kathol and Weihed, 2005). The late to post Svecokarelian granites likely originated from magmas derived from the middle and upper parts of the crust during a period of intense deformation and regional metamorphism at approximately 1.82-1.76 Ga (Kathol and Weihed, 2005).





However, the exact tectonic evolution and timing of the magmatism in the area is still debated and a couple of different models exist (Rutland et al., 2001; Rutland. et al., 2001; Weihed, 2003; Juhlin et al., 2002; Lahtinen et al., 2009; Skyttä et al., 2012).

Similarly, the age of the BSZ is still discussed and suggestions range from 1.825 Ga (Romer and Nisca, 1995) to 1.86 Ga (Rutland et al., 2001; Rutland. et al., 2001; Skiöld and Rutland, 2006) and 1.895 Ga (Weihed et al., 2002). However, most
authors agree that the peak of regional metamorphism in the area was around 1.825 Ga (Rutland et al., 2001; Rutland. et al., 2001; Weihed et al., 2002) and no major re-activation has so far been documented after 1.79 Ga. Since no borehole or seismic data are available, the geometry of the BSZ at depth is interpreted from surface observations and tectonic considerations. Romer and Nisca (1995) suggest a vertical strike-slip zone whereas Rutland et al. (2001) prefer a south side up dip-slip system.

The Burträsk fault scarp consists of a series of mostly southwest-northeast oriented segments, together forming a c. 35 km
long lineament (Fig. 2). In the easternmost part, it follows a deformation line belonging to the BSZ and in the central part, it continues sub-parallel to the BSZ, cutting through a large intrusion east of Bygdeträsk (Fig. 2). In the westernmost part, the deformation zones of the BSZ change direction to a more east-west orientation and the fault scarp starts to diverge from the BSZ (Fig. 2). The fault scarp is usually 5–10 m high and generally covered by a variable layer of Quarternary sediments dominated by till, clay and silt. In a few locations, scarp outcrops forming slightly overhanging cliffs are observed (Lagerbäck
and Sundh, 2008). Based on sediment liquefaction features close to Umeå, Mörner (2005) suggested a M>7 event for the formation of the fault scarp. On the other hand, Lagerbäck and Sundh (2008) found several sediment liquefaction and water escape structures during an extensive study of sediment deformation in the area, but were not able to establish any relationship between the intensity of deformation and the proximity to the fault scarp.

## 3   The local crossdip correction

### 3.1   Previous applications of the crossdip correction

One of the main challenges of crooked acquisition lines is that the trace midpoints have an offset component perpendicular to the profile direction. In the following, we will refer to this component as 'cross-offset'. As a consequence, the depth to a reflector with a crossdip component can vary for individual traces in a CDP gather, leading to an additional term in the traveltime equation (Nedimović and West, 2003):

$$t^2(x,y,h) = (t_0(x) + p_y y)^2 + p^2 h^2 \qquad (1)$$

where $t$ is the traveltime; $t_0$ is the zero time of the reflection; $x$ is the inline offset; $y$ is the cross-offset; $h$ is the source-receiver distance; $p$ is the slowness in profile direction and $p_y$ the slowness perpendicular to the profile.

These time shifts are not accounted for in the standard NMO processing which can result in focusing problems. Larner et al. (1979) defined the crossdip correction $\Delta_{tcross}$ in the form of:

$$\Delta_{tcross} = \frac{2sin\phi}{v}y = p_y y \qquad (2)$$

where $\phi$ is the crossdip angle and $v$ is the velocity.



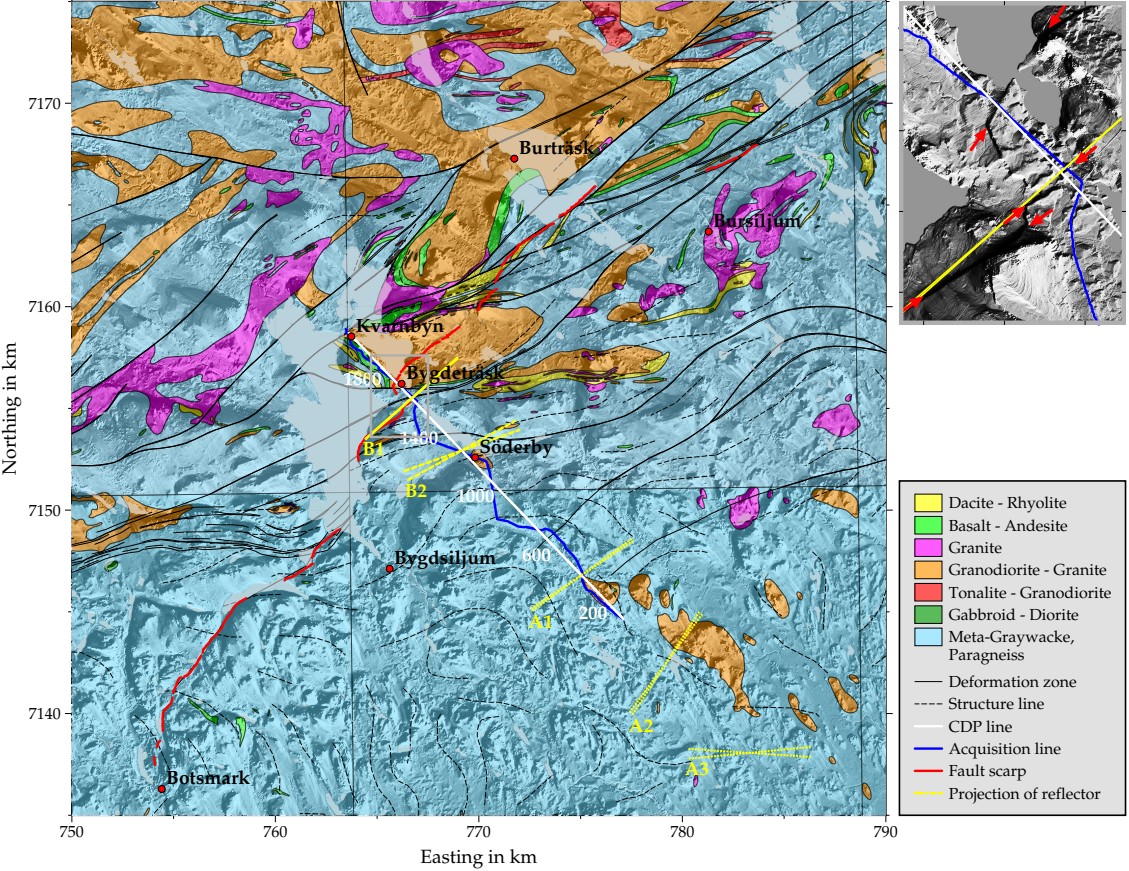

**Figure 2.** Geological map of the survey area and location of the Burträsk profile. The yellow lines mark the surface projection of reflector planes corresponding to the most prominent reflections in the seismic data with solid lines used for well constrained surface locations and dotted lines for poorly constrained locations. Note that the actual reflection points are shifted laterally for reflectors with a large crossdip component. The map inset shows an enlargement of the fault scarp and the red arrows indicate the location of the fault scarp. The location of the map is indicated in Fig. 1. Geological map ©Geological Survey of Sweden; elevation data ©Lantmäteriet.

The underlying assumption of this correction is that the reflector has no dip component in the profile direction (see Nedimović and West (2003) for a more general form of the crossdip correction).

The effect of uncorrected crossdip on the stacked seismic section depends mainly on the distribution of the cross-offset, i.e. on the geometry of the acquisition line. Figure 3 shows a synthetic example of an NMO corrected gather including two reflections affected by crossdip at 0.4 s and 1.2 s, respectively. The cross-offset distributions of these reflections correspond to the distributions at CDP 350 and 1350 of the Burträsk profile (Fig. 2). The energy of the upper reflection (A) is completely smeared and is hardly visible in the stack. For the lower reflection (B), one cross-offset value is dominating the whole dis-





tribution, causing the energy to focus at 0.8 s instead of 1.2 s. As a result, the reflection appears in the stack, but shifted in time.

Unlike the case for a reflector dipping in the profile direction, the crossdip correction has no lateral component, but it requires specific knowledge of both crossdip angle and velocity in the cross-profile direction or cross-slowness, respectively. Since these

parameters are usually unknown, they have to be derived from the data.

Despite the simple form of Eq. 2, the correction has been applied in quite different ways by different authors. Initially, Larner et al. (1979) calculated the crossdip correction simultaneously with the residual statics solution. In their fundamental paper, Nedimović and West (2003) developed a procedure for an automated crossdip correction where they invert for the cross-slowness by a grid search using the product of semblance and a local running average of the amplitudes as the objective

function. For each estimated cross-slowness, they evaluate the reliability by thresholding the stack's amplitude and modal filtering. Finally, they correct for the estimated crossdip while stacking using a slant stack approach. Other authors have used the same slant stack approach, but with manually determined crossdip values (Kim and Moon, 1992; Kim et al., 2014). A more simplistic approach is applying the crossdip correction as a static shift to the whole trace (Lundberg and Juhlin, 2011; Ahmadi et al., 2014). However, this approach generally decreases the quality of all other reflections in the trace and is consequently

mostly used for analyzing the crossdips without applying the correction (Urosevic et al., 2007; Rodriguez-Tablante et al., 2007; Malehmir et al., 2009; Dehghannejad et al., 2010, 2012; Hedin, 2015).

While the problems with applying the crossdip correction as a static shift are obvious, there are some more subtle disadvantages with implementing it using the slant stack approach. First of all, the slant stack procedure makes any further processing after the crossdip correction impossible. This might be problematic since there are very likely interactions between crossdip,

DMO and NMO corrections. Another issue occurs when a CDP gather is dominated by one cross-offset value. In this case, the reflection occurs twice in the stack: at the origin time as well as at the shifted time corresponding to the dominating cross-offset (red arrows in Fig. 3c). To avoid this reflection duplication, we decided to use a method which moves the energy of a crossdipping reflection to its origin time. Furthermore, we opted for a manual analysis of the crossdip angles since automatic detection as in the method of Nedimović and West (2003) is susceptible to noise and might pick up inline dip variations and

NMO residuals.

## 3.2 A new module for local crossdip correction

We developed a module for a local crossdip correction of individual reflections that can be directly used in a commercial software package. In our module, the reflections are approximated by a polygonal chain with the crossdip values defined at the vertices of the chain. Between the vertices, the crossdip values are linearly interpolated. For the actual correction, we use a

cut-and-shift method where reflections affected by crossdip are cut in a window around the biased reflection time $t_0 + \Delta t_{cross}$, shifted back by the correction term $\Delta t_{cross}$ and added to the trace at their origin time $t_0$ (Fig. 3). This procedure can lead to gaps in the corrected trace, but is necessary to prevent reflection duplication as in the case of reflection B in Fig. 3c.

Along with the crossdip values, the user defines the width of the correction window at each vertex and a constant velocity for the whole profile. We have decided to use the crossdip angle as the main parameter since it is more intuitive than the cross-



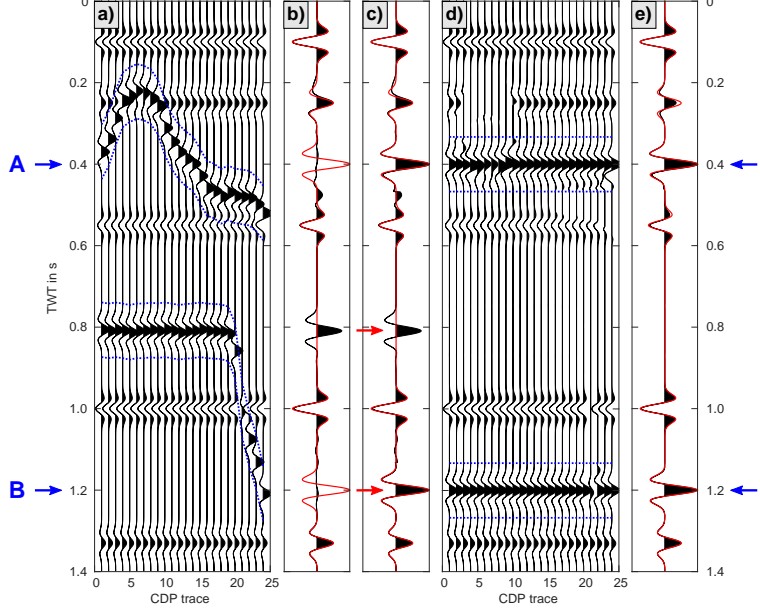

**Figure 3.** Synthetic CDP gather illustrating the principle of the crossdip correction. Panel (a) shows a NMO corrected CDP gather including two reflections with 10°crossdip. The cross-offset distribution corresponds to the cross-offset distribution of CDPs 350 and 1350 of the Burträsk data. The comparison of the uncorrected stack (wiggle trace) with the theoretical stack (red line) in panel (b) indicates that reflection A is missing and reflection B is shifted. In the slant stack (c), reflection A is correctly imaged but reflection B appears twice in the trace. After crossdip correction, the energy is aligned at the correct time in the CDP gather (d) and in the corrected stack (e), both reflections are correctly imaged.

slowness, but it is very important to be aware of the coupling between crossdip angles and velocities. Therefore, we recommend to translate variations and uncertainties in the velocity into a range of possible crossdip angles for geological interpretation.

The module also includes an interactive function for analyzing the crossdip angles. Analogous to velocity analysis, the crossdip values can be picked interactively on panels corrected with a constant crossdip angle. Thereby, it is possible to observe

5 the effect of the correction on a whole reflection and not just on a CDP gather.

## 4 Synthetic tests

We created a series of synthetic test datasets using the same acquisition geometry as in the Burträsk dataset. The modeling is based on the simple modeling approach outlined by Ayarza et al. (2000). In this method, the traveltimes are calculated from the geometry of the acquisition line and the reflectors, while the amplitudes are obtained using the formulas of Aki and Richards

10 (1980). The seismic traces are created by convolution of a scaled spike at the calculated traveltime with a Ricker wavelet.



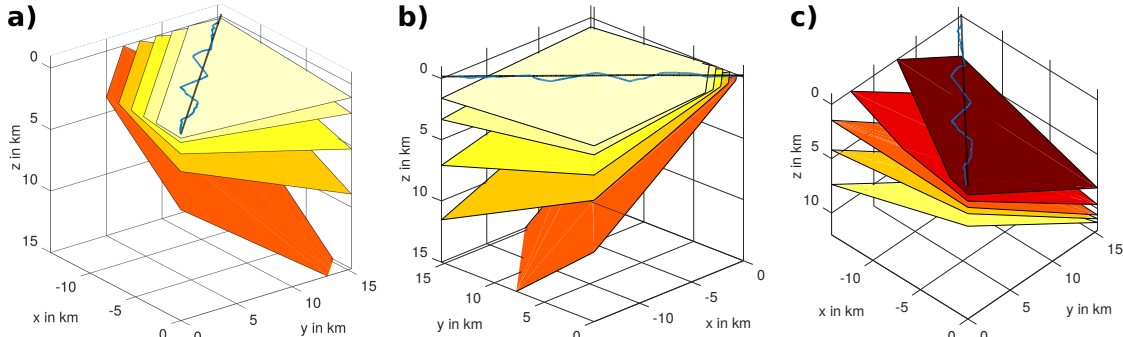

**Figure 4.** Reflector geometry used in the synthetic modeling. Model 1 (a) comprises reflectors with 5°, 10°, 20°, 30° and 45° crossline dip from top to bottom. Model 2 (b) consists of a series of reflectors with 5°, 10°, 20°, 30° and 45° inline dip. Model 3 (c) features reflectors with inline dips of 8.5°, 16.1°, 22.2°, 26.6° and 29.1° and crossline dips of 29.1°, 26.6°, 22.2°, 16.1° and 8.5°. The blue line represents the acquisition line and the black line the CDP line of the Burträsk dataset.

## 4.1 Model 1

The first model consists of a series of reflectors with increasing crossdip in a constant velocity medium (Fig. 4a). The objectives of this model were to test the correction method and to analyze the effects of crossdip on the stacked section. Figure 5a shows a stack of the synthetic data from model 1 using the true model velocities. Depending on the geometry of the acquisition line and the dip angle, the reflector crossdip manifests itself as interference effects and smearing of the reflections in the stacked section. A subsequent analysis of the optimum stacking velocities yielded alternating high-/low-velocity patches mimicking the mean cross-offset. However, the stack quality remained poor for the reflections with larger crossdips since the NMO correction can only account for hyperbolic traveltime distortions (Fig. 5b). When applying our crossdip correction module, we were able to retrieve the correct crossdip angles for all reflectors and the reflections in the stacked section were effectively focused and flattened (Fig. 5c).

### 4.1.1 Model 2

The second model was set up to evaluate how much inline dip can be picked up by the crossdip correction and consisted of a series of plane reflectors with increasing inline dip in a constant velocity medium (Fig. 4b). As expected, it was not possible to find any crossdip angles that consistently improved the stack of this dataset. Applying the crossdip correction clearly distorts the reflections (Fig. 6). Some very localized focusing also occurs, but overall, the smearing of the reflections increases considerably.



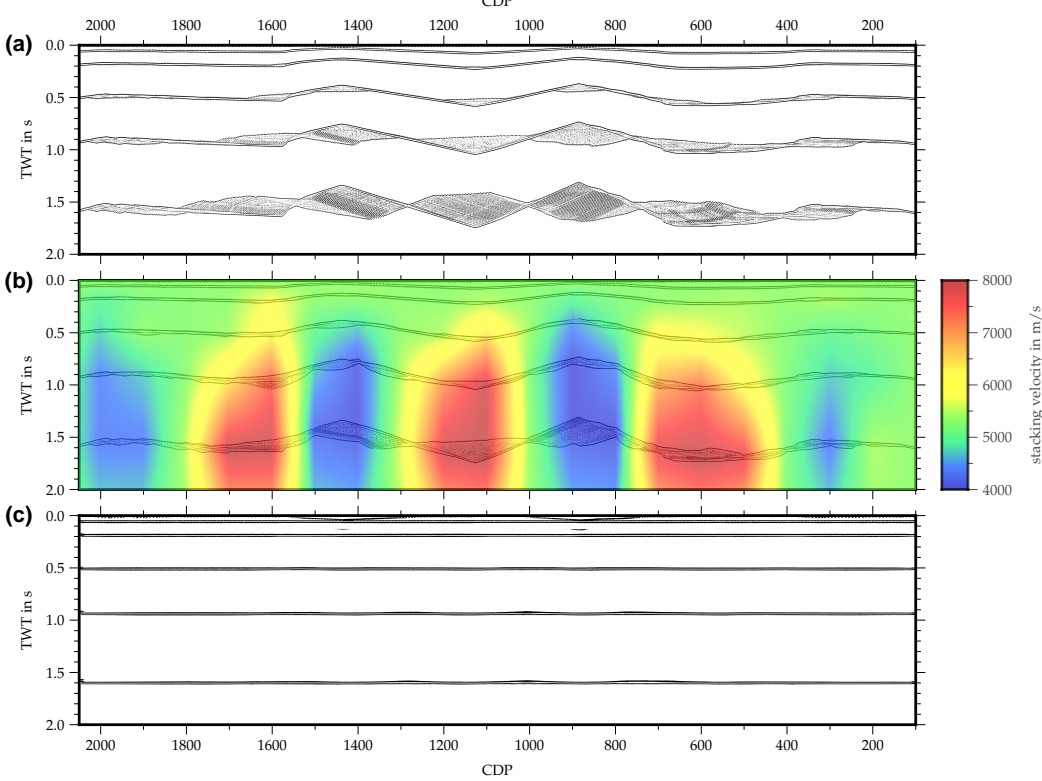

**Figure 5.** Results of the first synthetic model with reflectors dipping in the crossline direction. The reflections in the stacked section without crossdip correction (a) are heavily deformed and smeared. Using the optimum stacking velocities instead of the model velocity (b) reduces the smearing locally, but the the reflections remain distorted. After crossdip correction, the reflections become focused and flattened (c).

## 4.2 Model 3

The aim of the third model was to test the interactions of the crossdip correction and the DMO correction and to establish the preferred order of these steps in the processing flow. The model features a series of reflectors with different ratios of inline and crossline dip in a constant velocity medium (Fig. 4c). Figure 7 shows two versions of the stacked section with (a) crossdip

5  correction applied before DMO correction and (b) after DMO correction. In the first case, picking the optimum crossdip angle proved to be a bit challenging due to local maxima caused by the uncorrected inline dip. However, these occurred within a few degrees of the correct value and we were able to retrieve the expected values when picking with a focus on consistency. In the second case, similar difficulties arose because of artifacts introduced by the DMO correction. Again, focusing on consistency helped to identify the optimum crossdip angles. In both versions of the stack, the reflections become flattened and focused, but

10  the first version, with the crossdip correction applied before the DMO correction, contains less artifacts and is slightly more coherent.




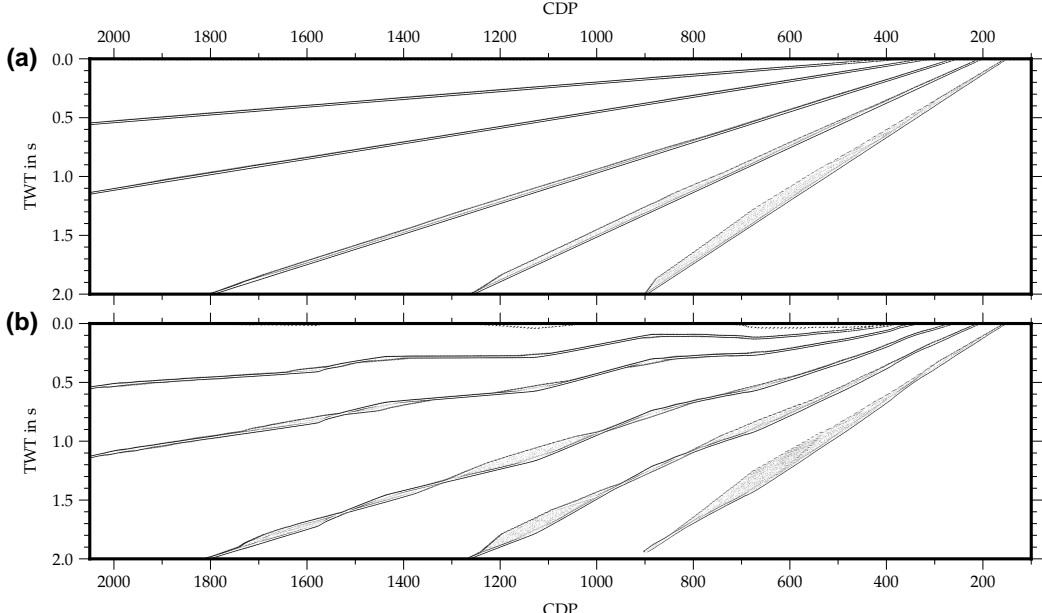

**Figure 6.** Results of the second synthetic model with reflectors dipping in the inline direction. The top panel (a) shows a stack without crossdip correction that exhibits the normal smearing due to inline dip. The bottom panel (b) shows a stack with a constant crossdip correction of -10° applied. The reflections get clearly distorted due to the falsely applied crossdip correction. In most areas, smearing is increased but locally, reflections focus slightly.

### 4.3 Implications for the implementation of the crossdip correction

As demonstrated clearly by the results of the first model, crossdip effects can be picked up by stacking velocities, manifesting themselves as alternating high-/low-velocity patches. Therefore, it is important to re-analyze stacking velocities after correcting for crossdip and to review the obtained crossdip angles with the updated velocity model. Furthermore, the results from the

5    second and third model highlight the importance of picking only crossdip angles that consistently improve the image of a whole reflection (segment) in order to exclude local optima caused by interactions between the effects of inline and crossline dip. The advantage of the manual crossdip correction approach is that it is possible – to a certain extent – to identify and avoid these interactions whereas the automatically conducted DMO correction is inevitably influenced by them.

Based on these results, we recommend an iterative processing sequence, where the crossdip correction is applied after an

10    initial velocity analysis, followed by repeated velocity analyses and DMO correction. As mentioned above, the crossdip angles should be reviewed after this sequence and the whole sequence should be repeated, if necessary. These recommendations are only valid for a manual crossdip correction. If the correction is derived automatically from the data, DMO should be applied first to avoid picking up local effects of inline dip.

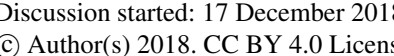



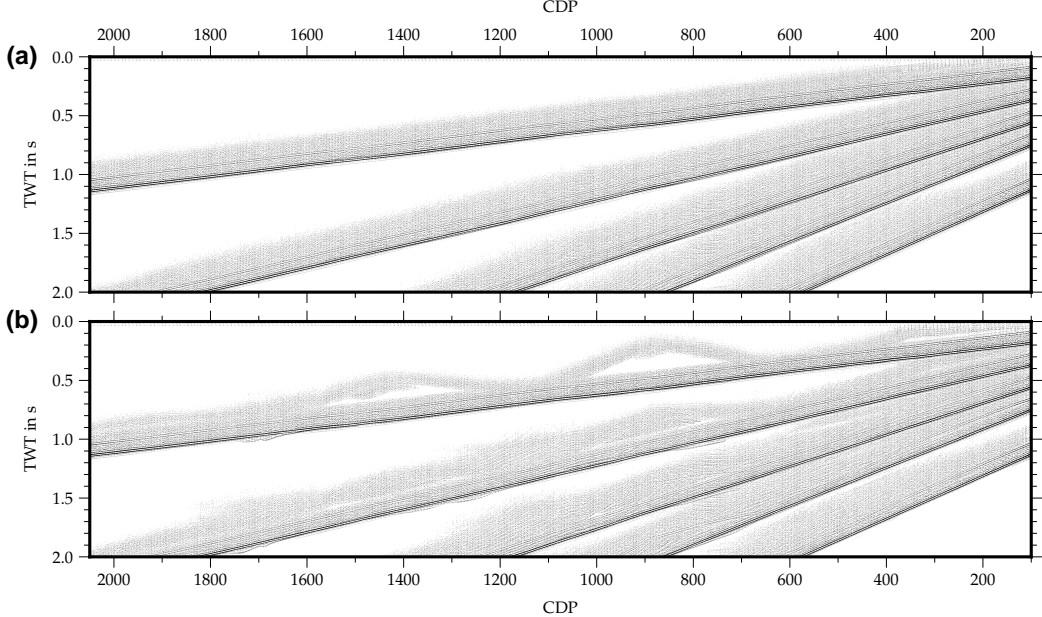

**Figure 7.** Results of the third synthetic model with reflectors dipping in the inline and crossline direction. Comparison of the stacked section with crossdip correction before DMO correction (a) and with crossdip correction applied after DMO correction (b). In both cases, the reflections become focused and flattened but in the second case, artifacts are stronger.

## 5    Reprocessing of the Burträsk dataset

### 5.1    Data Processing

The Burträsk profile was recorded on 280 channels with 20 m spaced 28 Hz single component geophones. The signal was generated using a VIBSIST hydraulic hammer source (Cosma and Enescu, 2001) with a nominal shot spacing of 20 m. Since the source could not be activated in the vicinity of buildings, shot coverage is quite sparse close to inhabited areas. As a result, the fold varies considerably over the profile (Fig. 9a). Similarly, the data quality is affected by random noise and coherent noise originating e.g. from the source, surface waves and ground roll to a varying extent. All shot gathers have a relatively high background noise level, but some feature very distinct first arrivals and clearly visible reflections whereas others show mostly noise with first arrivals barely identifiable after a few hundred meters (Fig 8). As a first order estimate of the average signal-to-noise ratio, Fig. 9a illustrates the ratio between the mean amplitude of the background noise in a 200 ms window before the first arrival and the mean amplitude of the first arrival in a 200 ms window starting at the first arrival (see also Fig 8).

The first part of the reprocessing followed the original processing flow from Juhlin and Lund (2011) quite closely. Due to the high noise level in some areas, we re-picked the first arrivals manually and re-analyzed the stacking velocities. We applied an additional ground roll and first break muting step following Oren and Nowack (2018). In their method, ground roll and



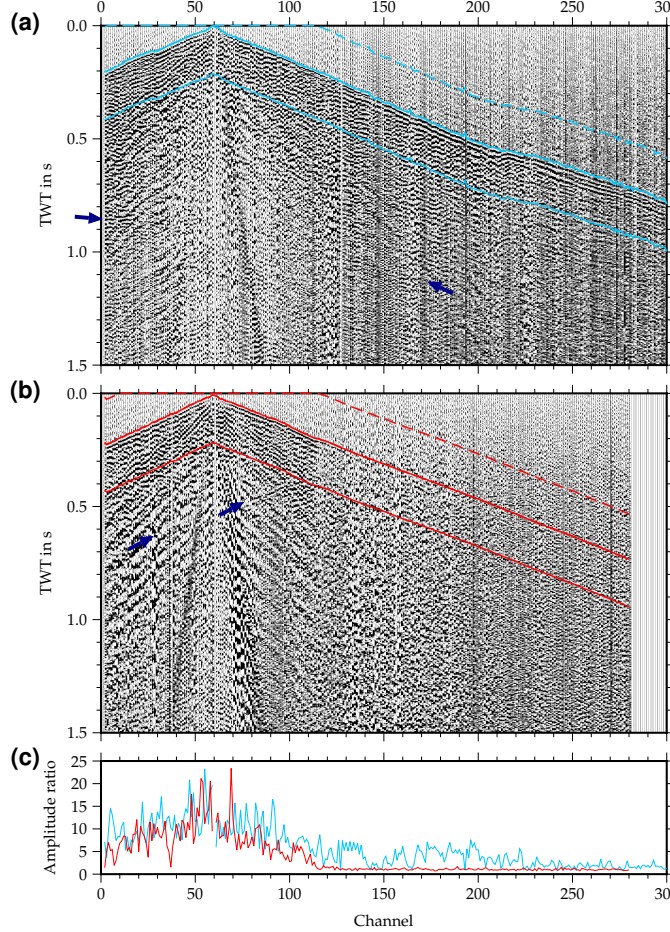

**Figure 8.** Unprocessed shot gathers illustrating variations in the background noise level. The upper shotgather (a) displays very clear first arrivals along the whole spread whereas the lower shotgather (b) is dominated by noise for larger offsets. The light blue and red lines mark the windows used for estimating the amplitude ratio between background noise and first arrival (c). The dark blue arrows indicate reflections.

first breaks are estimated by soft thresholding in the local time-frequency domain (Liu and Fomel, 2013) and subtracted from the data. Apart from that, there are only minor differences to the original processing, including slightly lower bandpass filter frequencies and a spherical divergence correction instead of an automatic gain control. A summary of the processing is given in Table 1.

5    Following the procedure outlined in the Sect. 4.3, we incorporated the crossdip correction directly after residual statics corrections, re-analyzed the velocities, applied a DMO correction and analyzed the velocities again. With the new pre-DMO velocity model, we updated the crossdip angles and re-applied the DMO correction using the post-DMO velocity model. During the first pass of this procedure, the velocities changed substantially after each step, but converged during the second pass. The final stacking velocity model is much more consistent along individual corrections than the initial one, but seems to be still





biased by dip effects. For comparison, we also tested applying the crossdip correction after the DMO correction, but could not produce a stack of comparable clarity.

Migration testing using a smoothed stacking velocity model for migration yielded poor results, confirming that the stacking velocities do not correspond to true subsurface velocities. The best results were archived for a Stolt migration with a constant velocity of 5400 ms$^{-1}$. This velocity is also consistent with the velocities obtained during the refraction statics correction. Finally, the section was depth converted using the same constant velocity. However, it is important to note that the depth values are only an estimation of the real depth since considerable uncertainties exist in the deeper part, where the velocity is poorly constrained.

Additional to the reflection seismic processing, we carried out first break tomography using the *PStomo_eq* solver by Tryggvason et al. (2002, 2009). We carried out the inversion in 3D with a cell size of $(x, y, z) = (20\,\mathrm{m}, 20\,\mathrm{m}, 10\,\mathrm{m})$ while simultaneously estimating a statics solution to eliminate the influence of sub-grid scale velocity variations on the model.

## 5.2 Results

Figure 9 shows a comparison between the unmigrated original stack, a reprocessed version of the stack excluding the crossdip correction and a reprocessed version including the crossdip correction. The original stack contains a series of northwest dipping reflections and reflection packages, referred to as A1-A4, and a series of southeast dipping reflections, named B1-B3. Even without the crossdip correction, the reprocessing enhanced the continuity and coherency of the reflections considerably. Especially reflections A3, B3 and B4, which are barely visible in the original stack, are imaged much clearer after reprocessing.

Analysis of the crossdip angles yielded mostly values in the range of $\pm\,16°$except for two deeper reflection packages, both with a crossdip of 24°(Fig. 9d). Northwest of CDP 1600, cross-offsets proved to be too small to determine any crossdip angles. The effect of the crossdip correction on the individual reflections depends to a large extent on their respective cross-offset distribution and crossdip angle. Not surprisingly, there is little change for the mainly inline oriented reflections A1 and B1 (Fig. 9). Reflections A2 and A3, which are in an area with comparably small cross-offsets, become more focused and can be followed to greater depth after crossdip correction, but stay essentially in the same position. In contrast to this, reflections B2 and B3 have vertical shifts of up to 100 ms caused by large, unevenly distributed cross-offsets. As a results of these shifts, reflection B2 looses its slightly listric appearance (Fig. 9). Moreover, two reflections appear only after crossdip correction: a rather weak reflection directly above reflection package B3 and a sub-horizontal reflection package close to the lower end of reflection B2, called C1 in the following (Fig. 9d).

Before converting inline and crossdip angles into strike and dip of the reflection, the possible range of velocities should be translated into a possible range of crossdip values. Table 2 shows inferred strike and dip values for a $\pm15\%$ velocity variation, corresponding to a velocity range of 4590 ms$^{-1}$ to 6210 ms$^{-1}$. Note that this is not an error estimate since it does not include uncertainties in the picking of the crossdip values.

None of the above mentioned reflections are associated to the segment of the fault scarp intersecting the seismic line at CDP 1720. There are some indications of dipping reflections which might be connected to the fault scarp, but these are too weak to interpret with confidence (Fig. 10). However, the surface projection of reflection B1 coincides with the extrapolation



**Table 1.** Summary of the processing flow used for reprocessing the Burträsk dataset.

| Step | Parameters |
|---|---|
| 1 | Manual firstbreak picking |
| 2 | Trace balance: *0-3000 ms* |
| 3 | Groundroll and firstbreak muting in local time-frequency domain: *25% threshold* |
| 4 | Spectral equalization: *30 Hz window, 25-40-120-150 Hz bandpass* |
| 5 | Time variant bandpass filtering: |
| | *0-200 ms: 35-60-120-180 Hz* |
| | *250-500 ms: 30-50-120-180 Hz* |
| | *600-900 ms: 25-40-110-165 Hz* |
| | *1100-3000 ms: 20-35-100-150 Hz* |
| 6 | Refraction statics: *floating datum, replacement velocity from model* |
| 7 | Trace editing |
| 8 | Horizontal median filter: *11 traces, 5300 $ms^{-1}$ & 3000 $ms^{-1}$* |
| 9 | Butterworth filter: *20-40-90-120 Hz* |
| 10 | Spherical divergence correction: *0.8 tpower, 2.0 vpower* |
| 11 | Velocity analysis |
| 12 | NMO correction: *40% stretch mute* |
| 13 | Residual statics |
| 14 | Crossdip correction: *5400 $ms^{-1}$, 20% taper*; |
| | Velocity Analysis |
| 15 | DMO correction; |
| | Velocity Analysis |
| 16 | Stacking |
| 17 | FX Deconvolution: *19 trace window* |
| 18 | Trace balance |
| 19 | Stolt migration: *5400 $ms^{-1}$, 0.6 stretch factor* |
| 20 | Zeromute |
| 21 | Approximate depth conversion: *5400 $ms^{-1}$* |

of the scarp segment west of the seismic line around CDP 1600 (Fig. 2). At the same location, the geological map features a deformation zone, but the strike does not agree well with the estimated strike of reflection B1 (Fig. 2).

Figure 11 shows the velocity model from the first break tomography. Due to the sparse spatial coverage, we reduced the 3D model to a 2D image displaying the mean of the model within 100 m distance from the receiver line in bright colors and otherwise the average of the model in the crossline direction in pale colors. Due to the high velocity contrast between the



**Table 2.** Estimated strike and dip the most prominent reflections.

| Reflector | Inline dip | Crossdip range assuming ±15% velocity variation | Inferred strike | Inferred dip |
|---|---|---|---|---|
| A1 | $13°$ | $2° \pm 0.3°$ | $234° - 237°$ | $\sim 13°$ |
| A2 | $26°$ | $-6° \pm 0.9°$ | $213° - 217°$ | $26° - 27°$ |
| A3 | $17°$ | $16° \pm 2.5°$ | $265° - 274°$ | $22° - 25°$ |
| B1 | $49°$ | $-2° \pm 0.3°$ | $48° - 49°$ | $\sim 49°$ |
| B2 | 34-40° | $-14° \pm 2.2°$ | $61° - 70°$ | $36° - 41°$ |

Reflection B2 branches in the upper part, therefore we used two different inline dip values in the estimation.

Quaternary sediments and the bedrock, most rays were guided along the bedrock surface and did not penetrate into the deeper parts.

In the upper part of the bedrock, the velocity is mainly around 5300–5500 ms$^{-1}$ and increases slightly with depth (Fig. 11). The model features several low velocity zones; the largest one at around $x = 4.0$ km to $x = 4.5$ km and coinciding with the location of the fault scarp, some smaller ones around $x = 5.0$ km, $x = 5.5$ km and $x = 9.0$ km and a couple of very localized ones throughout the whole profile.

## 6 Discussion and Interpretation

### 6.1 Crossdip correction

Both synthetic modeling and the field data example have clearly demonstrated the benefits of applying the crossdip correction to crooked line seismic data. As several previous studies have already illustrated, crossdip can de-focus and smear reflections, resulting in a poor stacked image (e.g. Larner et al., 1979; Kim and Moon, 1992; Nedimović and West, 2003). Another quite rarely mentioned aspect is that it can lead to vertical shifts, distortion and duplication of out-of-plane reflections, as well. Our local crossdip correction addresses this problem by shifting back reflections affected by crossdip and thereby projecting them into the CDP plane. However, the drawback of this procedure is that it can not handle crossing reflections very well. Therefore, the applicability of our method to areas with complex, distributed reflectivity patterns is limited. In such areas, more extensive testing is needed to develop an appropriate correction scheme since existing schemes, like the one of Nedimović and West (2003), do not account for the reflection duplication problem.

Apart from improving the imaging of crooked line seismic data, the crossdip correction has the advantage of extracting information on the 3D orientation of reflections, information which can be crucial for the geological interpretation of the seismic data. This can be especially important in low signal-to-noise data with low fold where sparse swath 3D processing is not an option.





## 6.2 Data Reprocessing

Processing crooked line seismic data from hardrock settings includes a whole range of different challenges. Apart from crossdip effects, imaging quality is often affected by residual static shifts, surface waves, coherent noise, strong ground roll, etc. As in many other studies (e.g. Juhlin, 1995; Pretorius et al., 2003; Urosevic et al., 2007; Place and Malehmir, 2016), the most

important steps in the basic reprocessing of the data proved to be the manual picking of first arrivals for improving the static correction and a careful velocity analysis. Furthermore, the comparison between Fig. 9b and 9c illustrates that the final image benefited considerably from using an analytical gain instead of applying an AGC since the contrast between the main reflections and the background reflectivity is preserved.

## 6.3 Origin of the reflections

The reflections in the final stack are mostly planar and occur in a relatively low reflectivity surrounding (Fig. 10). There are different possible origins for such reflections. The first possibility is a contact between different lithological units. Along the CDP line, the surface geology map features several lithological contacts (Fig. 2 & 10), but these contacts will only produce reflections if there is a significant difference in seismic impedance between the two lithological units. Both the output bedrock velocity from the refraction statics correction and the first break traveltime tomography do not show any significance changes

in bedrock velocity along the profile. This observation along with the lack of correlation between the seismic reflections and surface geology (Fig. 10) and the fact that some reflections are clearly cross-cutting sub-horizontal background reflections (Fig. 10), make lithological contacts a rather unlikely candidate. A second possibility is deformation or shear zones with either decreased or increased seismic impedance due to fracturing and/or re-mineralization processes. Whether or not such zones are visible in seismic data depends not only on the impedance contrasts, but also on the width of the zones. Theoretically, features

with a minimum width of $\lambda/30 - \lambda/20$ (Sheriff and Geldart, 1995), corresponding to 2.4–4.5 m for a peak frequency of 60–75 Hz and a velocity of 5400 ms$^{-1}$ in this survey, are detectable in seismic reflection images. In practice, the detectability limit depends strongly on the signal-to-noise ratio of the data. In the Burträsk survey, the signal level is relatively low, so a more realistic estimation of the detectability limit is $\lambda/12 - \lambda/8$, corresponding to a width of 6–11.25 m. Since the profile intersects a couple of deformation zones belonging to the BSZ (Fig. 2), it is quite plausible that some of the reflections are caused

by shear zones. Another possible scenario is that some reflections originate from magmatic dykes and/or sills. The Burträsk area was subject to intense migmatization and hosts intrusions from different magmatic pulses prior to, during and after the Svecokarelian orogeny (Kathol and Weihed, 2005). Thus, both sills and dykes are likely to occur along the profile.

In the following, we will discuss the nature of the most important reflections in the Burträsk profile. B1 is a relatively weak planar reflection that can only be observed clearly over a short depth interval, but in the unmigrated stack there are

some indications that it might continue to greater depths. (Fig. 9) Moreover, it seems to cut through the mostly sub-horizontal background reflectivity (Fig. 10), arguing against a lithological boundary. Since the surface projection of reflection B1 coincides both with a projection of the western scarp segment (Fig. 2) and a low velocity zone in the tomography model (Fig. 11), we interpret B1 as a reflection from either the western scarp segment itself, or from the continuation of the shear zone along which




the western scarp segment has ruptured. The small vertical extent of the reflection might be explained by a rather narrow shear zone that drops below the detection limit as the frequency content and signal strength decrease with depth.

Similar to B1, reflection B2 cross-cuts sub-horizontal background reflections (Fig. 10) and its surface projection coincides with a narrow low velocity zone in the tomography model (Fig. 11). Again, we interpret B2 as a reflection from a shear zone,

but the relation to the Burträsk fault is less obvious. The prominence of the reflection might suggest that the movement of the post-glacial fault at depth took place along reflection B2 and that the visible fault scarp segments are merely the branches of that fault where the surface rupture occurred. Branching of post-glacial fault scarps seems to be a common phenomenon and has has been observed for the Pärvie and Lansjärv faults (Talbot, 1986; Juhlin et al., 2010; Ahmadi et al., 2015). Since the Burträsk fault is seismologically very active, earthquake locations might give a hint which fault plane is active at depth. Recent

studies show the micro-earthquakes clustering along a southeast dipping plane, but unfortunately, the accuracy of the locations is not sufficient to distinguish between the closely spaced reflections B1 and B2 (Lund et al., 2016). In any case, B2's estimated strike of $61°$–$70°$ is not consistent with the strike of the fault scarp but rather matching the trend of the BSZ in the southern part (Fig. 2). Therefore, we prefer to interpret B2 as a reflection from a local shear zone belonging to the BSZ and not connected to the Burträsk fault. It is, however, still possible that B2 extends further to the southwest and connects with the westernmost

scarp segment.

Since reflection B4 projects to the surface close to a mapped deformation zone (Fig. 2), we tentatively interpret it as another local shear zone belonging to the BSZ.

Both of the two deepest reflections, B3 and C1, exhibit strong reflectivity and terminate very abruptly at the upper end. Neither data fold and amplitude ratio (Fig. 9a), nor the overall impression of the image quality (Fig. 9d & 10), indicate a

significant drop in data quality, so the abrupt terminations seem to be real features. Together with the high reflectivity, this suggests that reflections B3 and C1 are caused by sill or dike intrusions.

Unlike most of the other reflections, A2 and A3 are dipping to the northwest and their approximate surface projections are well south of the BSZ (Fig. 2). At 0.8 s, reflection A2 is clearly intersecting with a sub-horizontal reflection segment (Fig. 9d & 10), precluding the possibility of a lithological contact. The geometry of reflections B2, A2 and A3 and the apparently lower

dip of reflection A3 could be interpreted as a positive flower structure, consistent with the proposed oblique convergence of the Skellefteå district from the southeast (Bergman-Weihed, 2001). However, the true dips of reflections A2 and A3 are relatively small and very similar (Table 2) and the estimated strikes of reflections A2, A3 and B2 do not match at all (Fig. 2). So in this case, the additional information from the crossdip analysis argue against the hypothesis of a flower structure. Instead, the occurrence of several magmatic bodies southeast of the profile points towards a magmatic origin of the reflections, possibly as

feeder dikes following pre-existing weak zones in the upper crust. It is not clear from the present data set how reflection A1 should be interpreted.

Even though only reflections B1 and B4 can be directly correlated to deformation zones in the geological map, the dominance of southeast dipping reflections in the northwestern part of the profile suggests that the BSZ formed as a south-side up dip-slip system as interpreted by Rutland et al. (2001) and not as a vertical strike-slip zone as suggested by Romer and Nisca (1995).

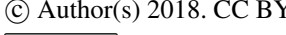



## 6.4 Imaging of the fault scarp

The lack of a clear reflection connected to the fault scarp at CDP 1710 be due to several different reasons. First of all, it could be caused by the absence of a resolvable contrast in physical properties. As discussed above, the fault zone needs to have a certain minimum width to be detectable in the seismic data. However, the post-glacial Pärvie and Suasselkä faults have successfully been imaged with reflection seismic using very similar acquisition parameters (Juhlin et al., 2010; Abdi et al., 2015, respectively). So if the Burträsk fault has similar characteristics, it should be well detectable in the seismic data. Another possibility is that the reflections from the fault scarp are not stacked properly due to the complex, three-dimensional geometry of the fault at the profile location. In this case, the reflections should still be clearly visible in the shot gathers. The shot gathers, however, exhibit only very blurred dipping reflections covered by various forms of noise. Figure 9a shows that between CDP 1600 and 1800, both the data fold and the average amplitude ratio between first arrival and the background noise are exceptionally low, indicating a poor signal-to-noise ratio. Therefore, the most likely explanation for the lack of a reflection connected to the fault scarp is simply the result of insufficient data coverage and quality.

## 6.5 Relation between post-glacial fault and BSZ

The relation between the Burträsk fault and the BSZ is still not fully understood. Compared to the majority of the known post-glacial faults, which predominantly strike north-northeast, the Burträsk fault has an unusually strong east-west component. In contrast to this, the neighboring Röjnoret fault is mostly north-south oriented, following yet another set of Paleoproterozoic shear zones (Fig. 1). This divergence from the main trend of post-glacial faults might indicate that the faults in the Skellefte area were to a very large extent guided by pre-existing weak zones in the crust. However, the Burträsk fault only follows individual deformation zones closely in the northernmost part and runs sub-parallel to the BSZ in the central part. Since the reflection seismic image has shown that there are potentially many more shear zones than the ones marked in the geological map, it is likely that the fault still follows weakness zones belonging to the BSZ. South of the large jump north of Bygdsiljum (Fig. 2), the fault scarp starts to diverge significantly from the BSZ where the latter turns to a more east-west orientation, suggesting that the orientation of the BSZ was no longer conducive to the prevailing stress field controlling the rupture direction. Therefore, we speculate that the BSZ acted as a guide for the Burträsk fault, causing its orientation to deviate from the orientation of the majority of post-glacial faults. During the rupture, the fault probably jumped between different weak zones to accommodate differences between the orientation of the BSZ and the orientation of the minimum horizontal stress.

## 7 Conclusions

In the first part of this paper, we presented a new software module for a local crossdip correction and tested the influence of crossdip on synthetic seismic data. An often forgotten effect of crossdip is that – depending on the cross-offset distribution – it can not only de-focus and smear reflections, but also shift them in time. Most existing crossdip routines rely on a slant stack approach for the correction which has the drawback that shifted reflections will appear twice in the stack. Within our




new module, we use a shift and stack method to overcome this reflection duplicating problem and enable further processing after the correction. The synthetic test examples demonstrate that the crossdip correction can interact both with the stacking velocities and the DMO correction. Based on these tests, we proposed an iterative processing scheme where a sequence of crossdip correction, velocity analysis, DMO correction and velocity analysis is repeated until the stacked image converges.

In the second part of this paper, we presented results of reprocessing data from the Burträsk profile using our new module. After crossdip correction, several reflections became significantly more continuous and coherent. An improved static solution and the use of an analytical gain also contributed considerably to the quality of the final image. The improvements we achieved during reprocessing illustrate the often underestimated potential of relatively simple methods, like the crossdip correction and the traditional filtering and muting, in processing noisy hardrock seismic datasets. Moreover, strike and dip values estimated

from the crossdip angles helped in associating the seismic reflections to geological features. We interpreted most of the southeast dipping reflections as shear zones belonging to the BSZ, implying that the BSZ is not a vertical, but a southeast dipping feature. The north to northwest dipping reflections in the southernmost part of the profile are likely attributed to magmatic intrusions. Due to the low data fold and high noise level close to the Burträsk fault, the scarp segment intersecting with the profile could not be imaged. However, we obtained a clear reflection from another scarp segment slightly further west, dipping

southeast at approximately $49°$.

*Author contributions.* C.J. and R.B. conceived the analysis method. R.B. developed the software module and carried out the data analysis and interpretation. R.B. prepared the manuscript and C.J. reviewed it.

*Competing interests.* The authors declare that they have no conflict of interest.

*Acknowledgements.* GLOBE *Claritas*[TM] under license from the Institute of Geological and Nuclear Sciences Limited, Lower Hutt, New

Zealand was used to process the seismic data. The Generic Mapping Tools package (GMT) was used to prepare many of the figures.



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



**Figure 9.** Comparison between the original processing of the Burträsk dataset and the reprocessing at different stages: (a) Fold distribution along the profile and mean amplitude ratio between the 200 ms of data and the background noise; (b) original stack; (c) reprocessed stack without crossdip correction; (d) reprocessed stack with crossdip correction overlain by the crossdip angles used in the correction. The small blue arrows indicate a possible continuation of reflection B1.



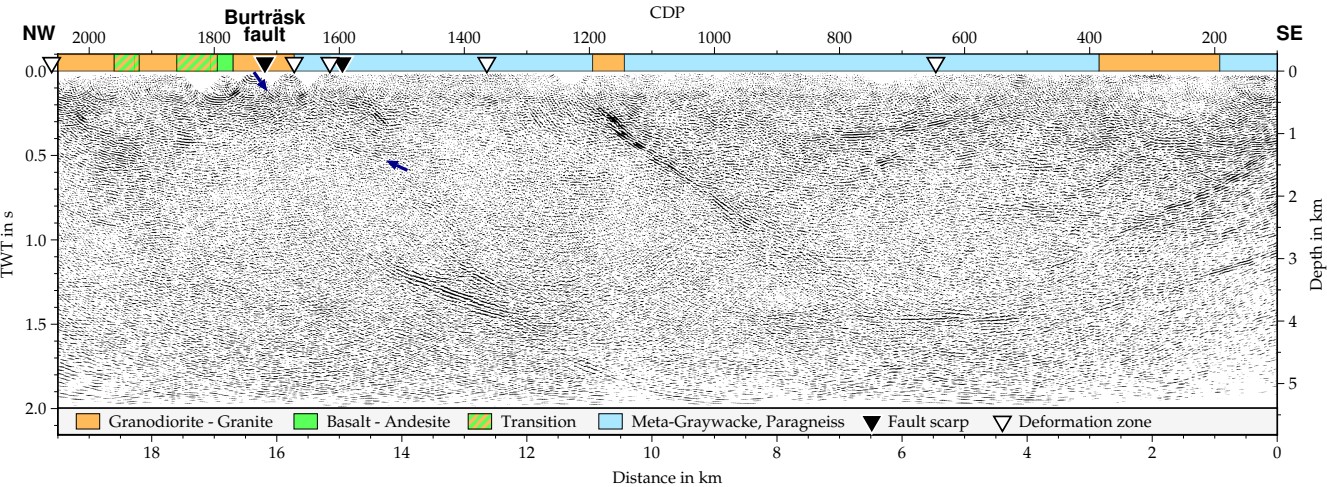

**Figure 10.** Migrated section along the Buträsk profile. For comparison, the surface geology is plotted on top of the section and the location of the fault scarp and deformation zones are marked. The small blue arrows highlight a very weak reflection that might be associated to the fault scarp. The vertical exaggeration of the section is approximately 1.

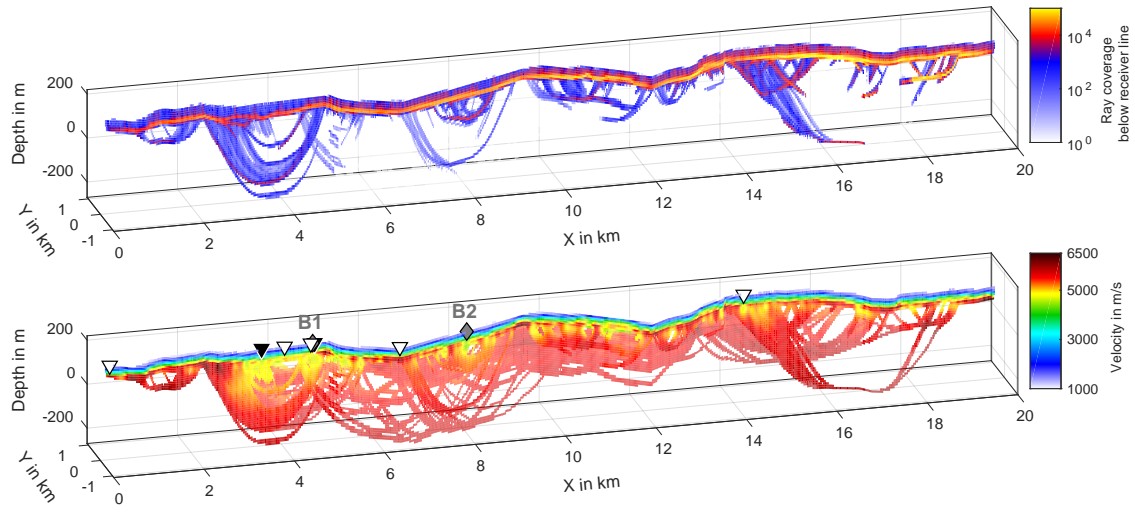

**Figure 11.** 2D slice of the ray coverage and the velocity model from the first break tomography. The upper panel shows the ray coverage in a 100 m wide zone below the receiver line. The lower panel displays the velocity below the receiver line in bright colors. In areas without velocity information below the receiver line, the average of the velocity model in the y direction is plotted in dim colors. The black triangles mark the location of the fault scarp, the white triangles indicate deformation zones and the grey diamonds correspond to the surface projection of reflections B1 and B2. The vertical exaggeration of the model section is 5.