# Peer review of "The crossdip correction as a tool to improve imaging of crooked line seismic data: A case study from the post-glacial Burträsk fault, Sweden"

_Solid Earth, 2018_

## Referee Comment (RC1) · Anonymous Referee #1 · 3 Feb 2019

This manuscript tackles the problem of obtaining reflector orientation from crooked line seismic data by so-called cross-dip correction. Standard method of cross-dip correction is revisited and implemented into a processing module. Subsequently, it is applied to the data from a post-glacial fault from N Sweden, improving significantly the stack quality and providing reflectors strikes and dips. It is a combination of the methodology and data application, which is well-written and easy to follow. There are only some minor issues that should be either supplemented or commented in the paper:

On Page 2, when discussing the attempts to extract 3D info from 2D crooked line

profiles, you can add reference to Wu et al. 1995 JGR paper (cross-dip), as well as Nedimovic and West (2003) part 2 paper on 3D pre-stack migration of 2D data + White and Malinowski (2012) paper on 3D post-stack migration of 2D data.

Page 2 line 19: what is so special about the post-glacial fault, that none of the cross-dip correction methods is suited for its imaging? Please expand or reformulate

Page 4 line 8: a south side up dip-slip system – please reformulate (e.g. using hangingwall)

Page 4 line 31: what is "crossdip angle"; which velocity "v" you are referring to? I think it might be instructive to make a cartoon, illustrating a crooked line acquisition over a reflector with inline/crossline dip

Page 6 – section 3.2: it's unclear what you are really implementing in your module; looks like it's the Nedimovic and West method just being applied manually? The description of your method can not be easily followed – please make maybe some more explicit link to Fig. 3 + change the caption of Fig. 3 accordingly

Synthetic tests: since there is an interplay between cross dip and velocity analysis I suggest to make a test with the model similar to model 3, but with different velocities between the reflectors.

Page 11 line 3: reformulate the sentence about the acquisition

Page 12 line 5-9: your workflow is not very clear: consider adding a cartoon illustrating it; Also, there is an interplay between residual statics and cross-dip correction, which you don't mention. Can you comment on that? Also, e.g. Wu et al. 1995 applied cross-dip correction first and then the residual statics.

Page 13 line 9-11: information on the refraction tomography is popping up here out of context; while I see your point later, when you show in Fig. 11 the low velocity zones close to the fault location, in my opinion this information could be omitted (and Fig. 11 as well). I'm sure that this velocity anomaly must be already present in your refraction

statics model + be visible in the first-breaks. If so, just add a comment. I don't like the results of the first-arrival tomography to be shown without the proper discussion on how it was performed, what is the resolution, checkerboard tests, etc.

Figure 2: lines are too thin + colors not well visible (e.g. acquisition line in blue)

Figure 9: is it a DMO stack? Just a comment: it's hard to determine to what extend the new results are influenced by the new reprocessing scheme or by cross-dip correction only. Even though the statics and velocities are linked together and you revised both of them, it might be worth to show the new reprocessing data stacked with the old velocities (and possibly statics).

---

## Referee Comment (RC2) · Anonymous Referee #2 · 4 Feb 2019

The contribution is interesting and it is worth pursuing its publication. The authors developed a tool which corrects crossdip effects in crooked seismic lines. The correction is applied to the data before stacking. In contrast to other methods, they use an iterative, manual procedure to overcome some disadvantages in the other methods. The method is demonstrated for synthetic data and to crooked line crossing the post-glacial Burträsk fault in Sweden, showing a significantly improved stacked image.

The manuscript is well-written, logically organized, and the figures are appropriate. To increase the value of the manuscript there a few things that I would suggest:

[Figure]

P 2 / L 18-19: " none of the existing correction methods is optimally suited to image a feature like a post-glacial fault" I would welcome some more discussion why these methods are not optimal in this specific case.

P4 / L10-13: It would be helpful to mark some of the mentioned aspects in Fig. 2. E.g. with A,B,...

P4 / eq. 1: use p_x instead of p for the inline slowness.

P 5 / L5: add A and B: "crossdip at 0.4 s (A) and 1.2 s (B)"

P 5 / L6: mark the CDP 350 and 1350 in Fig. 2.

P 5 / L7: "visible in the stack (Fig. 3b)."

P11: The reference to Fig. 9 appears before the reference to Fig. 8 in the text. This should be in order.

P13 / L3-8: Which types of migration were tested?

P13 / L9-11: What was the velocity model used for? Was migration also tested with this velocity model? How was the migration result using the tomography result, compared to the 5.4 km/s constant velocity?

P13 / L13-14: Add b c and d in the text.

P16 There is a discussion about the origin of the reflectivity. You are discussing about positive and negative impedance contrast which would mean either a mineralized, or a shear zone. Were the polarity and shape of the reflections analyzed? Are there any indications about impedance contrasts or e.g. tuning effects?

P17 / L8: "has has"

Fig 2: Some colors are hard to see (e.g. the gray box and the white numbers)

Fig 3b: Mark the shifted reflection B as you did it in Fig. 3c for the double reflection

Fig 9: Add A1 – 3 and B1 -4 also to c and d. This would make it easier to follow the descriptions in the text.

I think it would be illustrative to add a figure showing a CDP gather for the real data example: before and after crossdip correction and a comparison of the stacked sections (as for the synthetic model in Fig. 3).

It is not stated explicitly in the text, but I guess the module is written for GLOBE Claritas.

---

## Author Comment (AC1) · 5 Mar 2019

We would like to thank the anonymous referee for the constructive and helpful review. In the following, we will answer to the referee's comments and suggestions and describe adjustments made in our manuscript. The referee's comments are in italic font and our response is given in the intended blocks. Additionally, we will attach a new version of the manuscript with tracked changes as a supplement. Please note that this version of the manuscript contains changes suggested by both referees.

[Figure]

*P 2 / L 18-19: " none of the existing correction methods is optimally suited to image a feature like a post-glacial fault" I would welcome some more discussion why these methods are not optimal in this specific case.*

We explain this point in more detail later when we discuss the previous applications of the crossdip correction, but we fully agree that a short explanation is required here and we added this to the manuscript.

*P13 / L3-8: Which types of migration were tested?*

We tested a couple of different migration algorithms with different parameters including: Stolt migration with constant velocity and a stretching factor (best results), a phase shift migration with turning rays (very similar to the Stolt migration), Gazdags's phase shift migration with a 1D velocity model (quite strong artifacts), Kirchhoff migration using the smoothed stacking velocities (very smeared image, disrupted reflections) and FD migration with an interval velocity model derived from the stacking velocities (steeper reflections basically disappeared). We even tried the Kirchhoff and FD migration using a constant velocity model but the results were still not comparable to the results from the Stolt migration.
We added a short listing of the tested migration methods to the manuscript.

*P13 / L9-11: What was the velocity model used for? Was migration also tested with this velocity model? How was the migration result using the tomography result, compared to the 5.4 km/s constant velocity?*

We did not test using the velocity model from the tomography for migration since most rays did not penetrate the bedrock and therefore, the model only covers the top 50-200 m of the profile. Consequently, we would have to choose a velocity function for the deeper parts anyway. Directly below the surface, the tomography

model has of course a much higher resolution, but in this part, we have very poor data coverage due to groundroll muting. Since all migration algorithms that can handle 2D velocity models yielded blurred results, even for constant velocities, we decided to stick to the constant velocity Stolt migration.

Consequently, the tomography model was only used to get a better image of the velocity distribution in the shallow subsurface for comparison with potential shear zones.

*P16 There is a discussion about the origin of the reflectivity. You are discussing about positive and negative impedance contrast which would mean either a mineralizated, or a shear zone. Were the polarity and shape of the reflections analyzed? Are there any indications about impedance contrasts or e.g. tuning effects?*

The discussion of impedance contrasts and tuning effects is meant to point out potential geological structures that can cause the reflections. Unfortunately, the data quality does not allow a more detailed analysis of polarity and waveform of the reflections. The noise level is quite high in most parts of the profile and the source wavelet does not have a very impulsive nature – most likely due to interactions between the free surface and the very shallow sediment-bedrock contact. Even after deconvolution, the signal retains its ringy character. Therefore, a detailed analysis of the shape of the reflections would be, in our opinion, over-interpreting the data.

We have added a short comment about this to the manuscript.

*I think it would be illustrative to add a figure showing a CDP gather for the real data example: before and after crossdip correction and a comparison of the stacked sections (as for the synthetic model in Fig. 3).*

We agree that showing an example of a CDP gather before and after crossdip correction helps illustrating the correction. We have added one example including

a (comparably) very strong reflection, but we would like to stress that this example is **not representative** since most reflections are hardly identifiable in the CDP gathers and show up first clearly in the stack as coherent events contrasting with mostly uncoherent noise. This is likely similar in many hardrock environments and therefore, the crossdip analysis should be carried out on constant crossdip stack panels instead of CDP gathers.

Furthermore, we would like to thank the referee for pointing out the following, more technical issues in the manuscript. We implemented the suggested changes in the new version of the manuscript.

*P4 / L10-13: It would be helpful to mark some of the mentioned aspects in Fig. 2. E.g. with A,B,...*

*P 5 / L5: add A and B: "crossdip at 0.4 s (A) and 1.2 s (B)"*

*P 5 / L6: mark the CDP 350 and 1350 in Fig. 2.*

*P 5 / L7: "visible in the stack (Fig. 3b)."*

*P11: The reference to Fig. 9 appears before the reference to Fig. 8 in the text. This should be in order.*

*P13 / L13-14: Add b c and d in the text.*

*P17 / L8: "has has"*

*Fig 2: Some colors are hard to see (e.g. the gray box and the white numbers)*

*Fig 3b: Mark the shifted reflection B as you did it in Fig. 3c for the double reflection*

[Figure]

*Fig 9: Add A1 - 3 and B1 - 4 also to c and d. This would make it easier to follow the descriptions in the text.*

Please also note the supplement to this comment:
https://www.solid-earth-discuss.net/se-2018-120/se-2018-120-AC1-supplement.pdf

————————————————

---

## Author Comment (AC2) · 5 Mar 2019

We would like to thank the anonymous referee for the constructive and helpful review. In the following, we will answer to the referee's comments and suggestions and describe adjustments made in our manuscript. The referee's comments are in italic font and our response is given in the intended blocks. Additionally, we will attach a new version of the manuscript with tracked changes as a supplement. Please note that this version of the manuscript contains changes suggested by both referees.

[Figure]

*On Page 2, when discussing the attempts to extract 3D info from 2D crooked line profiles, you can add reference to Wu et al. 1995 JGR paper (cross-dip), as well as Nedimovic and West (2003) part 2 paper on 3D pre-stack migration of 2D data + White and Malinowski (2012) paper on 3D post-stack migration of 2D data.*

We thank the referee for pointing out the missing reference to Wu et al. (1995) and added it in the manuscript. However, we do not think that referencing the papers by Nedimović and West (2003) and White and Malinowski (2012) on 3D migration of crooked line data is very useful at this point since they do not deal with the crossdip correction and we do not build our research on the work presented in these papers.

*Page 2 line 19: what is so special about the post-glacial fault, that none of the cross-dip correction methods is suited for its imaging? Please expand or reformulate*

We explain this point in more detail later when we discuss the previous applications of the crossdip correction, but we totally agree that a short explanation is required here and have added this in the manuscript.

*Page 4 line 31: what is "crossdip angle"; which velocity "v" you are referring to? I think it might be instructive to make a cartoon, illustrating a crooked line acquisition over a reflector with inline/crossline dip*

The crossdip angle refers to the dip of a reflector perpendicular to the processing line and the velocity represents the medium velocity. As suggested, we have added an illustration of the ray geometry for a crooked acquisition line above a crossdipping reflector to the manuscript.

*Page 6 – section 3.2: it's unclear what you are really implementing in your module; looks like it's the Nedimovic and West method just being applied manually? The de-*

*scription of your method can not be easily followed – please make maybe some more explicit link to Fig. 3 + change the caption of Fig. 3 accordingly*

Our implementation of the crossdip correction differs significantly from the approach used by Nedimović and West. We manually determine the crossdip angle, calculate the theoretical crossdip correction $\Delta t_{cross}$ based on the the angle and shift back the data in a window around the reflection by $\Delta t_{cross}$ so that all data aligns at the origin time $t_0$. In contrast to this, Nedimović and West leave the data at its recorded position ($t_0 + \Delta t_{cross}$) and simply stack stack along their estimated crossdip correction time $\Delta t_{cross}$. As shown in the synthetic CDP gather (Fig. 3 in the manuscript), this slant stack approach can in some cases lead to a duplication of the reflection because the data are not actually shifted. In our approach, this can not happen.

We have attempted to explain this a bit more explicitly in the manuscript.

*Synthetic tests: since there is an interplay between cross dip and velocity analysis I suggest to make a test with the model similar to model 3, but with different velocities between the reflectors.*

The interplay between crossdip effects and stacking velocities occurs when the cross-offset distribution correlates to a certain extent with an NMO hyperbola. As a consequence, part of the crossdip effect can be picked up by the NMO correction, i.e. biases the stacking velocity model. Thus, the interplay is mainly related to the distribution of cross-offsets and not to the real underground velocities. Furthermore, the synthetic models are supposed to represent a typical hardrock setting where the velocities often remain approximately constant over very large areas. Therefore, we believe that we have already sufficiently illustrated and discussed the interplay between stacking velocities and crossdip effects in the second synthetic model and do not think adding another model will contribute significantly to our manuscript.

*Page 12 line 5-9: your workflow is not very clear: consider adding a cartoon illustrating it; Also, there is an interplay between residual statics and cross-dip correction, which you don't mention. Can you comment on that? Also, e.g. Wu et al. 1995 applied cross-dip correction first and then the residual statics.*

We acknowledge that our description of the workflow might have been a bit confusing, so we re-formulated the description both in sections 4.3 (implementation of the crossdip correction) and 5.1 (data processing) and added an illustration of our crossdip processing scheme.

We did not mention the residual static correction specifically in the description of the workflow, since we applied it as we would have done for any other dataset: We calculated the first pass after the initial NMO correction and tested a second pass after re-analysing the velocities. In our case, a second pass of residual statics did not lead to any significant improvements.

In the previous applications, the order between crossdip correction and residual statics was never consistent. Larner et al. (1979) calculated both simultane-aously, Kim and Moon (1992) applied them in a cascaded sequence, DuBois et al (1990) and Nedimović and West (2003) applied residual statics before the crossdip correction and Wu et al. (1992) and Rodriguez-Tablante et al. (2007) applied residual statics after the crossdip correction. This might be due to the fact that some authors, like Wu et al. (1992), have applied the crossdip correction as a static shift and others while stacking. Since we do not apply the crossdip correction as a static shift but as a local shift around specific reflections (depending on the cross-offset of the trace), we do not expect a large interplay with the surface consistent residual statics. In our opinion, it is best to remove residual static effects as far as possible before trying to correct for crossdip because this should improve the quality of all reflections and make them easier to track. However, we think that it is always worth to try to calculate a second, third, ... pass of residual statics as long as it improves the overall quality of the stack.

[Figure]

We added a short comment about this in the manuscript, as well.

*Page 13 line 9-11: information on the refraction tomography is popping up here out of context; while I see your point later, when you show in Fig. 11 the low velocity zones close to the fault location, in my opinion this information could be omitted (and Fig. 11 as well). I'm sure that this velocity anomaly must be already present in your refraction statics model + be visible in the first-breaks. If so, just add a comment. I don't like the results of the first-arrival tomography to be shown without the proper discussion on how it was performed, what is the resolution, checkerboard tests, etc.*

It is not possible to reproduce the low velocity zones with the refraction statics model because the refraction statics model is by design more sensitive to variations in the overburden and, more importantly, the model is two-dimensional to fit the line geometry. When testing the tomography, we also started with a 2D model, but had to switch to a 3D model to get any sensible results. However, we understand the concerns of the referee about just showing the results of a tomography. Since we think that the low velocity zones provide important constraints for our interpretation, we ran a checkerboard test to estimate the resolving power of our tomography. We perturbed the starting model with a 3D checkerboard pattern with a checker size of (x, y, z) = (500 m, 500 m, 100 m) and amplitude of 15 % and computed synthetic travel times for this model using a very fine slowness grid. For the inversion of the synthetic data, we used the same parameters as in the main inversion, but did not estimate a static solution. The ray paths of the checkerboard model are reasonably similar to the ray paths in the actual inversion. The results are shown in Figure 1. The uppermost checkers are reconstructed well but most of the lower checkers are poorly reconstructed or not reconstructed at all, indicating that only velocity variations close to the bedrock surface can be resolved properly and that the lower part of the tomographic model is poorly constrained. We expanded the description of the

tomography in the manuscript and added a summary of the checkerboard test. We hope that this addresses the concerns of the referee sufficiently.

*Figure 9: is it a DMO stack? Just a comment: it's hard to determine to what extend the new results are influenced by the new reprocessing scheme or by cross-dip correction only. Even though the statics and velocities are linked together and you revised both of them, it might be worth to show the new reprocessing data stacked with the old velocities (and possibly statics).*

We never intended to claim that all improvements are due to the crossdip correction. In Figure 9, we included both a reprocessed version of the stack without crossdip correction (c) and one with crossdip correction (d) to isolate the effects of both parts of the reprocessing. The comparison of the original stack and the reprocessed version without crossdip correction (both conventional DMO stacks) illustrates how the improved static corrections and velocity model lead to a significantly more coherent stack. The effects of the crossdip correction can be assessed by comparing the reprocessed stack without (c) and with crossdip correction (c). Just to clarify: both stack represent the best image we could get without and with crossdip correction and have been stacked with different velocity models. This is necessary since the stack is very sensitive to the choice of the stacking velocities and these are both affected by inline dip and crossdip. Thus, stacking the dataset with crossdip correction with the velocity model from the dataset without crossdip correction would seriously damage the stack quality and vice versa. Updating the velocity model is an integral part of our procedure to apply the crossdip correction and therefore we believe that separating them, as suggested, does not make sense. However, we now try to emphasize in the manuscript how the comparison of the different versions of the stack helps to isolate the effects of the conventional reprocessing and the effects of the crossdip correction.
Furthermore, we would like to thank the referee for pointing out the following issues in the manuscript. We have implemented the suggested changes in the new version of the manuscript.

*Page 4 line 8: a south side up dip-slip system – please reformulate (e.g. using hangingwall)*

*Page 11 line 3: reformulate the sentence about the acquisition*

*Figure 2: lines are too thin + colors not well visible (e.g. acquisition line in blue)*

Please also note the supplement to this comment:
https://www.solid-earth-discuss.net/se-2018-120/se-2018-120-AC2-supplement.pdf

———————————————————

**Fig. 1.** Checkerboard test to illustrate the resolving power of the tomography. From top to bottom: velocity distribution of the checkerboard model; relative velocity perturbation of the checkerboard model; re

**Supplement:**

[revised manuscript text omitted]